# Using the Cyclotide Scaffold for Targeting Biomolecular Interactions in Drug Development

**DOI:** 10.3390/molecules27196430

**Published:** 2022-09-29

**Authors:** Binu Jacob, Alicia Vogelaar, Enrique Cadenas, Julio A. Camarero

**Affiliations:** 1Department of Pharmacology and Pharmaceutical Sciences, University of Southern California, Los Angeles, CA 9033, USA; 2Norris Comprehensive Cancer Center, University of Southern California, Los Angeles, CA 9033, USA

**Keywords:** cyclotides, drug design, backbone cyclized polypeptides, protein–protein interactions, Cys-rich peptides

## Abstract

This review provides an overview of the properties of cyclotides and their potential for developing novel peptide-based therapeutics. The selective disruption of protein–protein interactions remains challenging, as the interacting surfaces are relatively large and flat. However, highly constrained polypeptide-based molecular frameworks with cell-permeability properties, such as the cyclotide scaffold, have shown great promise for targeting those biomolecular interactions. The use of molecular techniques, such as epitope grafting and molecular evolution employing the cyclotide scaffold, has shown to be highly effective for selecting bioactive cyclotides.

## 1. Introduction

Disruption of pharmacologically relevant protein–protein interactions (PPIs) remains a challenging task [1,2,3]. This is primarily due to the large and relatively flat binding surfaces involved in most PPIs. The most challenging molecular targets are those involving intracellular PPIs, which also require the therapeutic agent to cross the cell membrane efficiently [4,5].

Generally, we can consider two major structural types of therapeutic agents, small molecules and protein-based compounds, known as biologicals. Small molecules are small in molecular size (≤100 atoms) and usually show good pharmacological properties, such as cell permeability and stability. However, small molecules only provide a modest overall surface area available for interacting with the protein target. This has made quite challenging the identification of small molecules able to efficiently disrupt PPIs [6,7].

The use of polypeptide-based molecules, on the other hand, has provided efficient therapeutic tools to modulate PPIs with high specificity and selectivity [8]. For example, therapeutic monoclonal antibodies can target extracellular protein domains in a remarkably efficient fashion [9,10]. Antibodies, however, are expensive to produce, show low tissue penetration, are unable to reach intracellular targets, and cannot be delivered orally. These limitations have led to exploring alternative polypeptide-based scaffolds as a potential source of protein-based therapeutic leads [11,12,13,14,15,16,17,18].

The use of highly constrained polypeptides and able to cross membranes has recently received special attention for developing de novo stable polypeptide-based therapeutics [11,12,19]. Among the different highly-constrained peptide-based scaffolds, the cyclotide family has emerged as a fascinating family of medium-sized and Cys-rich plant-derived backbone-cyclized polypeptides (≈30–40 amino acids long). Cyclotides possess a stabilizing core formed by three disulfide bonds forming a Cys-knotted arrangement (Figure 1) [19]. This Cys-knotted backbone-cyclized (or circular) topology confers cyclotides with unusual characteristics such as remarkable stability to thermal/chemical denaturation and proteolytic degradation [20]. These unusual features have made cyclotides ideal tools for developing novel peptide-based therapeutic leads (see some recent reviews on the topic [11,21,22,23,24,25,26]. Due to their relatively small sizes, cyclotides can be chemically produced by standard solid-phase peptide synthesis (SPPS) methods and can also be produced by heterologous expression in different types of cells using standard expression vectors (see a recent review on the biological and chemical production of cyclotides [27]). Another intriguing property of cyclotides is that some of them can cross the cellular membranes of mammalian cells through endocytic mechanisms [28,29] and be able to modulate PPIs in vitro and in vivo [5]. Some naturally occurring cyclotides have also shown biological activity when given orally hence showing some oral bioavailability [21,30,31]. The first cyclotide discovered in plants, kalata B1 (Figure 1), was employed in traditional medicine as an effective uterotonic agent when given orally [20]. Other kalata B1-derived cyclotides have also been shown to possess biological activity when dosed orally [30,31].

These unusual properties have made the cyclotide scaffold an ideal molecular framework for producing novel peptide-based diagnostic and therapeutic agents by using molecular engineering and/or evolution strategies. This article provides an overview of their most relevant properties as well as their potential to be used as a molecular framework for the development of peptide-based therapeutic agents.

## 2. Structure

Cyclotides consist of a backbone-cyclized polypeptide with six Cys residues that form a Cys-knotted structure. Naturally-occurring cyclotides are medium-sized polypeptides containing from 27 to 37 amino acids (Figure 1), although larger engineered cyclotides have also been reported [5,33,34]. Loop 6 seems to be the most tolerant for the insertion of large sequences using molecular grafting techniques, tolerating the insertion of sequences from 14 to up to 25 residues [5,34]. This loop also seems to tolerate well the introduction of isopeptide bonds without affecting the folding and stability of the resulting engineered cyclotide [35].

As mentioned earlier, the six Cys residues on the cyclotide scaffold form three interlocked disulfides on a well-defined Cys-knot arrangement, with disulfide Cys^III^-Cys^VI^ running through the ladder arrangement formed by disulfides Cys^I^-Cys^IV^ and Cys^II^-Cys^V^ (Figure 1 and Figure 2A). This highly constrained topology, known as the cyclic cystine knot (CCK) motif, makes the cyclotide backbone extremely rigid and compact [36]. This explains their unusual stability to thermal/chemical denaturation as well as proteolytical degradation that is characteristic of naturally occurring as well as engineered cyclotides [37,38].

These properties were initially showcased on the first naturally occurring cyclotide, kalata B1, which was isolated and identified in the late 1960s by Gran while studying the traditional remedy employed by the indigenous people in central Africa use to accelerate childbirth [39]. This traditional medicine was prepared by boiling parts of the plant *Oldelandia affinis* (*Rubiaceae* family) to prepare the tea extract used as a remedy [40]. These early findings highlight the remarkable stability of the cyclotide scaffold that was biologically active even after being extracted by boiling water and providing uterotonic activity when dosed orally.

Since the discovery of the first cyclotide, more cyclotides have been isolated from other plant families [41]. Naturally occurring cyclotides have been mainly classified into three subfamilies, the Möbius, bracelet, and trypsin inhibitor cyclotide subfamilies [42]. Although all the cyclotides from the different subfamilies share the same CCK topology, they show differences in the size and sequence of the different loops (Figure 1**)**. An additional structural difference between the cyclotides from the Mobius and bracelet subfamilies is that Möbius cyclotides have a *cis*-Pro bond at loop 5 while bracelet cyclotides do not have it (Figure 2B) [33].

Even though bracelet cyclotides are by far the most abundant in nature, where they make up around 70% of all the know cyclotides thus far, they are very difficult to fold in vitro [43]. A recent report, however, has shown that introducing a single point mutation in loop 2, replacing Ile11 with either a Gly or Leu residue (Figure 1), can substantially increase the folding yield in bracelet cyclotides [44]. This was demonstrated in several bracelet cyclotides. This approach was successfully used to synthesize mirror image enantiomers and used quasi-racemic crystallography, allowing to elucidate of the first crystal structures of bracelet cyclotides containing an Ile residue in loop 2 [44]. This study should offer an alternative and efficient approach to obtaining bracelet cyclotides, facilitating easy access to their three-dimensional structures and providing a basis for further study of cyclotide structure and function and their future use as drug design scaffolds [44].

The trypsin inhibitor subfamily contains a relatively smaller number of cyclotides isolated from the seeds of several *Momordica spp.* plants (*Cucurbitaceae* family) [45,46,47]. These cyclotides do not share significant sequence homology with members from the Möbius and bracelet subfamilies beyond the CCK topology. As its name indicates, cyclotides from this subfamily are extremely potent trypsin inhibitors (*K*_i_ ≈ 20 pM) [48]. Cyclotides of this family show high sequence homology with squash trypsin inhibitors, which also contain a Cys-knotted structure, although they are not backbone cyclized, and sometimes are referred to as cyclic knottins [49].

New naturally occurring cyclotides with sequences rich in positively charged Lys residues have also been recently isolated from two plants from the *Violaceae* family in Australia (49). Unfortunately, so far, there is no information available on their chemical synthesis, making it difficult to evaluate their real potential as molecular frameworks in the design of novel peptide-based therapeutics.

## 3. Biosynthesis

Naturally occurring cyclotides are produced by enzymatic processing from ribosome-produced precursor proteins (Figure 3). Many of these precursors are encoded in genes containing multiple copies of the same or different cyclotide sequences [50]. For example, the first dedicated genes to produce cyclotides were isolated from the cyclotide-producing plant *Oldelandia affinis* (*Rubiaceae* family), which is the natural source for cyclotide kalata B1 [51,52]. The genome analysis in other cyclotide-producing plants from different families has also allowed the identification of similar genes involved in the bioproduction of cyclotides [19,46,53,54,55].

Asparaginyl endopeptidase (AEP)-like ligases have been shown to mediate the C-terminal cleavage and backbone-cyclization of the linear cyclotide precursor (Figure 3) [56,57], while papain-like cysteine proteases have also been found to participate in the N-terminal cleavage required for the AEP-mediated backbone cyclization [58]. Several AEP-like ligases have been shown to work in vitro, being able to cyclize different linear peptides, including cyclotide precursors polypeptides containing D-amino acids [56,59,60,61,62]. Protein-disulfide isomerases (PDIs) have also been shown to play an important role in the oxidative folding of cyclotides in vivo [63]. These findings open the exciting possibility of genetically engineered modified organisms for the bioproduction of cyclotides [64].

## 4. Chemical Synthesis

The chemical synthesis of cyclotides can be readily achieved using standard SPPS methods (for a detailed review on this topic, see [27]). Linear precursors can be easily produced by SPPS using Fmoc-based chemistry, the resulting linear precursors can be backbone-cyclized using an intramolecular version of native chemical ligation in aqueous buffers at physiological pH (pH ≈ 7) and then oxidatively folded sequentially (Figure 4A) [27]. A more convenient approach developed in our lab involves performing the cyclization and folding steps in a ‘single pot’ reaction which requires only using reduced glutathione (GSH) as a thiol additive during the native chemical ligation reaction [65]. Using this approach, we have generated many disulfide-contained backbone-cyclized polypeptides [66,67], including naturally occurring and engineered cyclotides [5,33,35,66].

Chemically-produced cyclotide linear precursors have been also chemoenzymatically cyclized using purified AEP-like ligases [57,58,63]. This cyclization approach does not require the linear precursor to be natively folded [56]. A similar approach has also been used to produce trypsin inhibitor cyclotides from the corresponding linear precursors linearized at loop 1 but using trypsin [68]. In this case, the linear precursor linearized between the residues at the P1–P1′ junction requires to be natively folded to be recognized by the enzyme trypsin, which then facilitates the cyclization reaction [68]. The introduction of mutations that disrupt the binding between the linear precursor and the enzyme trypsin has been shown to reduce the cyclization yield [27]. The transpeptidase-like sortase A (SrtA) has been also employed for the backbone-cyclization of cyclotide precursors [69]. However, it should be mentioned that SrtA-mediated cyclizations require a recognition-specific sequence in the ligation site that is only partially removed during the transpeptidation reaction leaving an extra heptapeptide at the cyclization site.

## 5. Recombinant Expression

Protein splicing units, also known as intein domains, can be used in cis or trans for the generation of backbone-cyclized polypeptides (more detailed reviews on this topic can be found at [27,70]). The first report of cyclotide production on a bacterial expression system used a modified *Mxe* GyrA intein designed to produce C-terminal a-thioester peptides that were cyclized in vitro using standard native chemical ligation conditions [71,72]. Naturally occurring split-inteins, which mediate protein *trans*-splicing (PTS), are by far more efficient in producing cyclotides, and they can be used in combination with both prokaryotic and eukaryotic heterologous expression systems (Figure 4B) [73,74,75]. The use of intein-mediated PTS allows the production of natively-folded cyclotides reaching intracellular concentrations that range from 20 to 40 µM [75]. For example, natively-folded cyclotide MCoTI-I has been expressed in bacterial cells yielding around 2 mg of folded cyclotide per 20 L of wet cells using an *Escherichia coli* expression system [75]. These values are similar to those found in cyclotide-producing plants, such as *O. affinis*, which has been reported to produce around 15 mg of folded cyclotide per 100 g of wet cells when grown under ideal conditions in vitro [76].

Hence, PTS provides an attractive and cost-effective alternative route to produce bioactive cyclotides by making use of bacterial expression systems. Other advantages include the availability of expression vectors, the fastest growth rate, and the simplicity of working with microorganisms.

The possibility to produce natively-folded cyclotides inside a living cell allows the generation of genetically encoded libraries of cyclotides that, when used in combination with bacterial expression systems, can reach library diversities of up to a billion different cyclotide sequences. These libraries can be readily screened in the intracellular milieu to quickly produce novel cyclotide sequences able to modulate/interfere with specific PPIs [74].

Heterologous expression of cyclotides also allows very low-cost access to cyclotides labeled with NMR active isotopes (^13^C and ^15^N), therefore making it possible to perform structural studies using heteronuclear NMR techniques [5]. These techniques have been recently used to elucidate the structure of a p53-activating bioactive cyclotide bound to its respective protein target (Figure 5A) [5].

## 6. Biological Activities of Naturally-Occurring Cyclotides

Cyclotides from the bracelet and Möbius subfamilies show insecticidal activities and are thought to work mainly as host-defense agents [19,51,77,78,79,80]. Other biological activities reported for cyclotides from these two subfamilies also include inhibiting the growth of mollusks [81], and nematodes and trematodes [82,83,84].

The biological mechanism of action for the cyclotides from these two subfamilies involves interacting with the cellular membranes of the gastrointestinal tract in insects disrupting its activity [85]. The molecular mechanism involves first the specific binding of the cyclotide to the phosphatidylethanolamine phospholipids present in the cell membrane. This series of binding events end up compromising the structure of the membrane facilitating the formation of pores and eventual cellular leakage of cytosolic components [86,87,88].

An interesting characteristic of cyclotides from these two subfamilies is that when natively folded show some amphipathic character due to the well-defined localization of hydrophobic and hydrophilic patches on the molecular surface of the folded cyclotide [88]. This molecular feature somehow mimics the amphipathic properties of some types of classical antimicrobial peptides. This molecular characteristic found in some cyclotides has been used to explain their antibacterial activity [89]. As an example, the Möbius cyclotide kalata B1 has been described to possess antimicrobial activity against Gram-positive and Gram-negative bacteria [90]. Other cyclotides isolated from the plants *Hedyota biflora* (*Rubiaceae* family) [91,92] and *Clitoria ternatea* (*Fabaceae* family) [55] have also shown similar antimicrobial activities. The naturally-occurring cyclotide with more potent antimicrobial activity tested so far is the bracelet cyclotide cycloviolacin O2 [93]. This cyclotide also showed activity in *Staphylococcus aureus* in a mouse infection model [94]. The antimicrobial activity of these types of cyclotides, when tested in vitro, has been shown to strongly depend on the buffer composition, showing good antimicrobial activity when low ionic buffers are used during in vitro testing. These findings may suggest that the in vivo antimicrobial activity of cycloviolacin O2 could be due to an indirect effect.

Cyclotide kalata B7, which was also isolated from the same plant where kalata B1 was also originally isolated, has been found to be a moderate agonist for G protein-coupled vasopressin V_1a_ and oxytocin receptors with EC_50_ values ranging from 1 to 10 µM [95]. Kalata B7 also interacts in the same way as kalata B1 does with membrane phospholipids inducing cellular toxicity, and it is very likely the cause of the hemolytic properties and cardiotoxicity observed in several cyclotides of the kalata family [96]. Accordingly, these types of cyclotides require further optimization before they can move forward as potential therapeutic leads.

## 7. Cyclotides with Novel Biological Activities

As indicated earlier, the unique features associated with the cyclotide scaffold make it an excellent molecular scaffold to be used for the generation of a novel class of polypeptide-based therapeutics (see Table 1) [11,12,25,26].

The highly compact structure provided by the CCK motif confers them with extreme resistance to chemical, physical, and proteolytical degradation. In addition, the loops decorating the Cys-knotted core on cyclotide loops 1 through 6 (Figure 1), are highly tolerant to the introduction of amino acid mutation as well as sequence insertions. This provides an ideal molecular scaffold for producing new cyclotides with novel biological functions using molecular grafting and evolution tools

Cyclotides from the trypsin inhibitory subfamily are excellent molecular platforms to design cyclotides with novel biological activities as they have been shown to cross cellular membranes and, in contrast with kalata cyclotides, are not cytotoxic to mammalian cells up to concentrations of 100 µM [5,29,30]. This allows them to target intracellular PPIs with minimal cytotoxic side effects [5].

One of the first examples of designing engineered cyclotides with novel biological activities involved the production of antiviral and anticancer cyclotides [70,99,100]. Angiogenesis is a well-validated target for developing anti-cancer agents. The cyclotide kalata B1 was used for the molecular grafting of several vascular endothelial growth factor A (VEGF-A) Arg-rich peptide inhibitors to produce cyclotides with anti-VEGF activity [112], with the most active cyclotide showing an IC_50_ value of 12 µM for the VEGF-A receptor. Similar grafting strategies have also been reported to generate kalata B1-based cyclotides able to inhibit melanocortin 4 and bradykinin receptors for pain and obesity management, respectively [30,110].

A point mutated kalata B1 cyclotide (T20K) also showed activity in a mouse model of multiple sclerosis when dosed orally [31]. Treatment at a dose of 20 mg/kg was able to impede disease progression without exhibiting adverse effects [31]. Although these studies did not provide detailed pharmacokinetic and/or pharmacodynamic data, they highlight their potential for providing orally active peptide-based therapeutics.

The cyclotides MCoTI-I/II from the trypsin inhibitory subfamily are the most used for producing novel cyclotides with new biological activities, which can be achieved by employing molecular grafting techniques (see Table 1). For example, cyclotide MCoTI-I was employed to produce potent CXCR4 antagonists [34]. This cytokine G-couple protein receptor (GPCR) is found to be overexpressed in many cancer cells, which is believed to drive tumor growth, neoangiogenesis, tumor metastasis, and cell survival [113]. CXCR4 cyclotide antagonists can also be used as bioimaging agents for visualizing cancer tumors expressing high levels of CXCR4 protein [114]. These results showcased for the first time the potential of cyclotides to be used as bioimaging agents. In this work, a [^64^Cu]-DOTA-labeled version of a CXCR4-binding MCoTI-based cyclotide was employed to detect tumors containing tumors cells with high-expression levels in mice in combination with positron emission tomography-computed tomography (PET-CT) [114].

Cyclotides from the trypsin inhibitor subfamily have also been successfully used to develop inhibitors for specific proteases. Many human diseases involve the dysregulation of specific proteases and therefore are well-recognized drug targets [115,116]. For example, sequence modification of loops 1 and 6 of cyclotide MCoTI-II produced a potent and selective foot-and-mouth-disease (FMDV) 3C protease inhibitor [68]. Similar approaches have been employed to design MCOTI-based inhibitors of human leukocyte elastase and b-tryptase, which are well-validated targets for inflammatory disorders [106,107]. More recently, a highly potent kallikrein-related peptidase 4 (KLK4) inhibitor (*K*_i_ ≈ 0.1 nM), which displayed 100,000-fold selectivity over related KLKs, was also reported by modifying loops 1 and 6 of cyclotide MCoTI-II [99].

The cyclotide MCoTI-II was modified by using molecular grafting on loops 1 and 6 to novel cyclotides able to inhibit the BCR-Abl kinase [103]. However, the level of inhibition obtained with these engineered cyclotides was rather modest, and they did not display significant activity in an imatinib-sensitive human chronic myeloid leukemia (CML) cell line.

MCoTI-grafted cyclotides were recently used to inhibit α-synuclein-induced cytotoxicity when expressed in baker’s yeast *Saccharomyces cerevisiae* [74]. The α-synuclein protein has been linked to Parkinson’s disease and therefore is a validated therapeutic target for Parkinson’s disease [117].

A recent study reported the first design and synthesis of a novel MCoTI-based cyclotide with broad-spectrum antimicrobial activity in vitro against different ESKAPE pathogens (*P. aeruginosa*, *S. aureus*, *K. pneumoniae*, and *E. coli*), including 20 clinical isolates for the human pathogens *P. aeruginosa* and *S.* aureus. The median minimal inhibitory concentration (MIC) 50% (MIC_50_) and MIC 90% (MIC_90_) values for several clinical strains of *P. aeruginosa* (*n* = 20) were 1.5 μM and 3.1 μM, respectively; while for clinical isolates of *S. aureus* (*n* = 20), the MIC_50_ and MIC_90_ were 6.25 and 12.5 μM, respectively [97]. The most active cyclotide also showed activity in vivo using a murine model of acute *P. aeruginosa* peritonitis [97]. These results demonstrate for the first time the design of an engineered cyclotide able to show potent antimicrobial activity in vitro using physiological-like conditions and, more importantly, in vivo, using a murine *P. aeruginosa*-induced peritonitis animal model, thereby providing a promising lead compound for the design of novel antibiotics.

One of the most exciting features found in some cyclotides is their ability to cross cellular membranes allowing them to target intracellular PPIs [28]. This was demonstrated for the first time in a report by engineering cyclotide MCoTI-I using molecular grafting into a potent antagonist for the interaction between the tumor suppressor protein p53 and the Hdm2/HdmX E3 ligase (Figure 5) [5]. This engineered cyclotide showed high cytotoxicity to several wild-type p53 cancer cell lines and was able to activate the p53 tumor suppressor pathway both in vitro and in vivo using an animal model of human colorectal carcinoma (Figure 5) [5]. This work highlights the ability of novel engineered cyclotides to target an intracellular PPI in vivo displaying the therapeutic potential of MCoTI-based cyclotides.

As discussed earlier, the development of biosynthetic approaches that allow the generation of natively-folded cyclotides inside living cells makes possible the production of large libraries of genetically encoded cyclotides that can be screened inside the living cell, thus preserving the link between phenotype and genotype. The expressions of such libraries using microorganisms, such as *E. coli*, should allow reaching complexities that could easily reach ≈ 10^9^ different cyclotide sequences. The production and screening of these cell-based large libraries of genetically encoded cyclotides should allow the use of selection approaches that mimic the molecular evolutionary processes found in nature.

This approach was first reported on the biosynthesis of a small library based on the cyclotide MCoTI-I where every residue, except those in loop 6 and those forming the Cys-knot, was mutated to evaluate the effects on folding and biological activity of the resulting mutants [118]. Most of the mutants were able to adopt a native cyclotide fold highlighting the high sequence tolerance of MCoTI-based cyclotides to mutations [118].

An acyclic version of the Möbius cyclotide kalata B1 was used in combination with a bacterial display approach to generate a genetically-encoded library that was screened for selecting novel cyclotide sequences able to bind the VEGFA binding site on neuropilin-1 [109]. This approach produced several bioactive cyclotides with a high affinity for neuropilin-1 that displayed in vitro activity inhibiting endothelial cell migration [109].

A linearized version of cyclotide MCoTI-II has been also employed as a scaffold to generate genetically-encoded libraries used in combination with a yeast-display system. This work produced several MCoTI-based cyclotides that bind to the cytotoxic T lymphocyte-associated antigen 4 (CTLA-4) [102]. CTLA-4 is expressed in T cells working as an immune checkpoint that downregulates the immune response and therefore is a validated target in the development of checkpoint inhibitors for cancer treatment [119].

A fully folded cyclotide-based genetically encoded library was recently used for phenotypic screening in eukaryotic cells [74]. In this work, a bioactive cyclotide able to reduce α-synuclein-induced cytotoxicity in baker’s yeast *S. cerevisiae* was readily selected by phenotypic screening from a library of cells transformed with a mixture of plasmids encoding active and inactive cyclotides in a ratio of 1 to 50,000 [74]. A similar approach was also used to generate a genetically encoded library based on the Cys-rich backbone-cyclized defensin RTD-1, which was screened for inhibitors against the anthrax lethal factor protease [120]. These results show the potential to carry out rapid phenotypic screening of genetically encoded cyclotide-based libraries in eukaryotic cells using activity-based rather than binding-based screening assays. The use of eukaryotic expression systems to produce cyclotide-based libraries also allows the generation of cyclotides with different post-translational modifications, which are not available in bacterial expression systems, therefore allowing to further increase the molecular diversity of the library.

More recently, the use of mRNA-display technologies has been successfully employed for the selection of potent cyclotide-based b-FXIIa inhibitors [98], although the pharmacology and activity of the most promising inhibitors were not investigated in animal models.

High-throughput screening of chemically-produced cyclotide libraries of cyclotides has been also made possible due to the accessibility of efficient methods for the chemical synthesis of cyclotides [65].

The use of a ‘tea-bag’ approach in combination with a high-efficiency “one-pot” cyclization-folding protocol allowed the rapid generation of a small amino acid scanning library based on a CXCR4 cyclotide antagonist [65]. Key to this approach was the inclusion of an efficient purification step to rapidly remove non-folded cyclotides from the cyclization-folding crude.

A similar approach was also recently used for developing inhibitors of the TNF-α converting enzyme (TACE) (*K*_i_ ≈ 150 nM) and anthrax lethal factor protease (*K*_i_ ≈ 40 nM) using the θ-defensin RTD-1 as a molecular scaffold [67].

## 8. Biodistribution Studies on Cyclotides

Recent reports have analyzed the biodistribution of cyclotides from the trypsin inhibitor subfamily and their ability to cross the blood-brain barrier [114,121]. They confirm that MCoTI-cyclotides distribute in mice mainly into the serum and kidneys, and they are predominantly eliminated through renal clearance [114,121]. It was also confirmed that the trypsin inhibitor cyclotide MCoTI-II cannot go across the blood-brain barrier [121]. However, it should be noted that biodistribution profiles of novel engineered cyclotides could be also affected by their new biological activities. A biodistribution study on a CXCR4 binding cyclotide in mice revealed major accumulation on the liver, spleen, and lungs even after 24 h of administration [114]. This cyclotide was also mostly secreted through renal clearance showing a peak after 90 min of administration and slowly decaying after 24 h.

Oral activity has been described for cyclotide-containing traditional remedies and demonstrated in several kalata B1-grafted cyclotides [31,39,40]. A recent study provided the first in vivo dose-exposure metrics for cyclotides using the prototypic cyclotide kalata B1 and two orally active kalata B1-grafted analogs, ckb-KAL and ckb-KIN [122]. This work determined the pharmacokinetic parameters in orally and intravenously dosed rats providing comparative pharmacokinetic parameters for natural and grafted cyclotides. The native and grafted cyclotides exhibited multiple compartment kinetics and measurable but limited oral bioavailability [122].

Preclinical pharmacokinetic evaluation is a crucial step in the progression of therapeutic candidates for use in a clinical setting, and more studies on bioactive cyclotides of other sub-families may be required. In addition, novel approaches to increase the oral bioavailability of cyclotides will have to be explored.

## 9. Summary

Cyclotides are now starting to become a well-studied family of micro-proteins that, based on their unique properties, are receiving acceptance as molecular scaffolds for the potential design of novel peptide-based therapeutics and diagnostic tools. Cyclotides possess extraordinary stability to thermal/chemical denaturation and proteolytic degradation due mostly to their CCK topology. Some cyclotides can cross cellular membranes, allowing them to target intracellular PPIs and, in the same case, are shown to be orally active [5,31,32]. Altogether, these features emphasize the high stability of the CCK topology to degradation–reduction under complex biological conditions.

The relatively small size of cyclotides allows their chemical synthesis by using standard solid-phase peptide synthesis methods. Chemical synthesis facilitates the introduction of chemical modifications, such as non-natural amino acids and PEGylation, to improve their pharmacological profiles [34,35].

The high plasticity and tolerance to sequence variation of cyclotides, combined with the ability to be produced using several heterologous expression systems, make the cyclotide scaffold an ideal substrate to perform molecular evolution techniques to select novel cyclotide sequences optimized to antagonize specific molecular targets [76,111,112].

Large-scale production of bioactive cyclotides can be easily accomplished by using different heterologous expression systems. The full characterization of AEP-like ligases and papain-like enzymes involved in the biosynthesis of cyclotides should also provide alternative routes for the generation of genetically-modified plants able to biosynthesize cyclotides.

## 10. Concluding Remarks

Disrupting PPIs in an effective and selective fashion remains a challenging task. This is mainly due to the nature of molecular surfaces involved in most PPIs, which are relatively large and flat. The cyclotide scaffold provides a highly constrained and stable molecular framework that can be efficiently used to antagonize extra- and intracellular PPIs.

The use of molecular grafting and molecular evolution techniques has been proven to provide cyclotides with novel biological activities. Novel cyclotides able to target many different molecular targets have been reported (see Table 1). Most of these cyclotides have been tested in vitro, although some were also shown to be active in animal models [5,31,32]. Some bioactive cyclotides have also been shown to be orally active [30,31], although more detailed studies on the oral bioavailability of cyclotides may be required.

Despite the numerous reports on bioactive cyclotides targeting pharmacologically-relevant PPIs, no cyclotides have reached human clinical trials yet. Some of the challenges affecting bioactive cyclotides (before they can move into the clinic) are the potential immunogenicity and oral bioavailability. However, it is anticipated that more studies on the pharmacological properties of these exciting new micro-proteins may be available very soon.

## Figures and Tables

**Figure 1 molecules-27-06430-f001:**
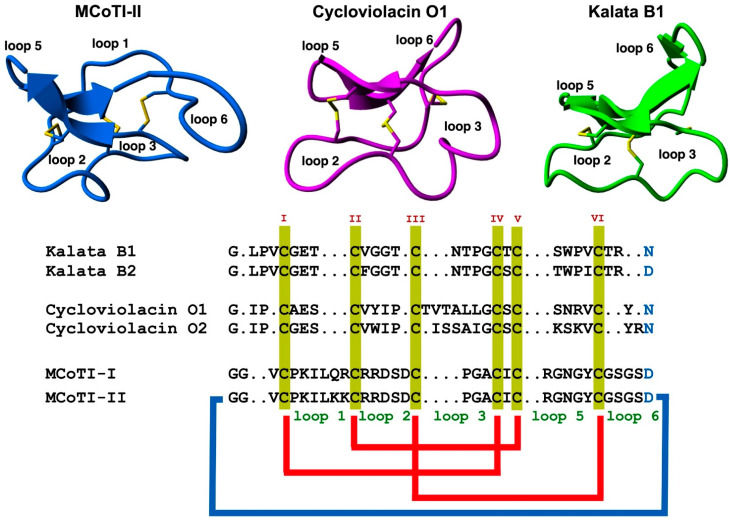
Structure and sequence alignment of cyclotides from the trypsin inhibitor (MCoTI-II, pdb: 1IB9) [32], bracelet (cycloviolacin O1, pdb: 1NBJ) [33], and Möbius (kalata B1, pdb: 1NB1) [33], subfamilies. Loops connecting the different Cys residues are designated with Arabic numerals, and the six Cys residues involved in the Cys-knot are labeled with roman numerals. Conserved Asp/Asn (required for backbone cyclization in nature) and Cys residues are marked in blue and yellow, respectively. Molecular graphics were created using Yasara (www.yasara.org) (accessed on 16 August 2022). Figure adapted from references [11,23].

**Figure 2 molecules-27-06430-f002:**
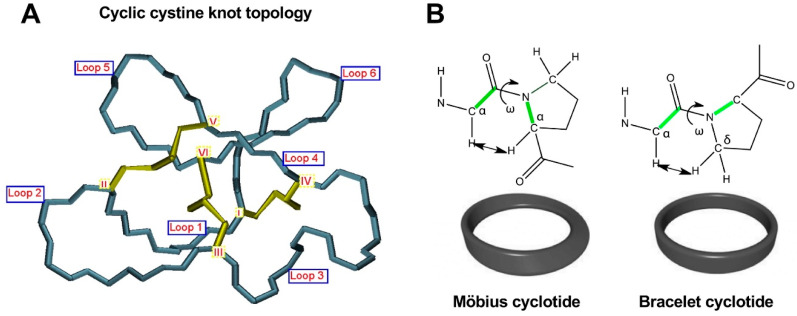
Detailed structural features of the cyclic cystine knot (CCK) topology observed in all naturally-occurring cyclotides. (**A**) Three-dimensional structure of the CCK architecture topology and the connecting loops found in cyclotides. Cys residues are labeled with roman numerals and loops connecting the different Cys residues are identified with Arabic numerals. (**B**) Cyclotides from the Möbius subfamily have a *cis*-Pro residue located in loop 5 that is responsible for inducing a local 180° backbone twist. This feature is absent in cyclotides from the other two subfamilies. Figure adapted from reference [26].

**Figure 3 molecules-27-06430-f003:**
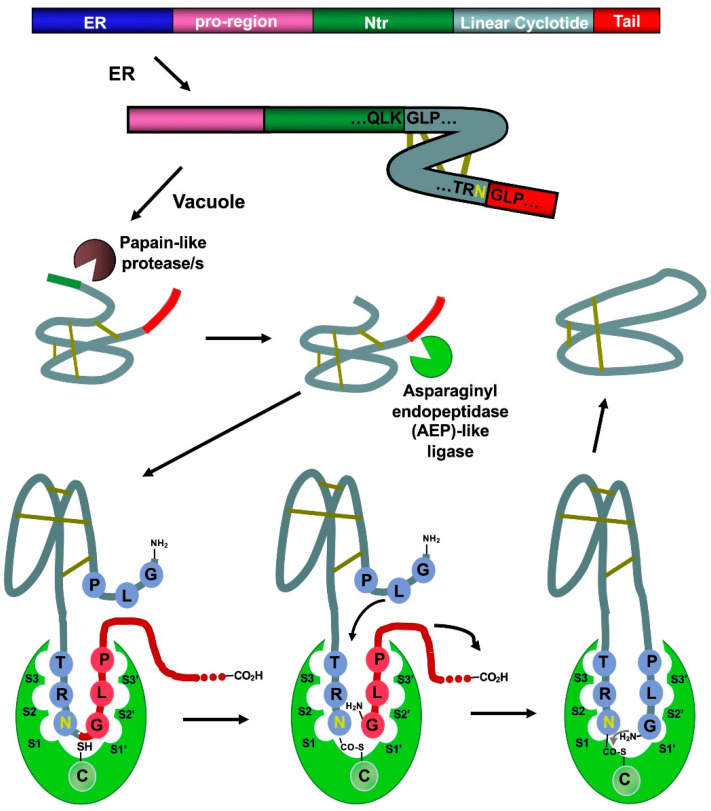
Scheme showing the proposed mechanism for the biosynthesis of cyclotide kalata B1. The cyclization step in cyclotides is mediated by an asparaginyl endopeptidase (AEP)-like ligase. The cyclization and cleavage of the C-terminal pro-peptide from the cyclotide precursor protein happen at the same time through a transpeptidation reaction, involving an acyl-transfer step from the acyl-ligase intermediate to the N-terminal residue of the cyclotide domain [50]. The protease responsible for the N-terminal cleavage required for the cyclization has been identified as a papain-like protease [58]. As shown in the scheme, the kalata B1 protein precursor contains an ER signal peptide, an N-terminal pro-region, the N-terminal repeat (NTR), the mature cyclotide domain, and a C-terminal flanking region (tail, also known as CTR). Figure adapted from reference [24].

**Figure 4 molecules-27-06430-f004:**
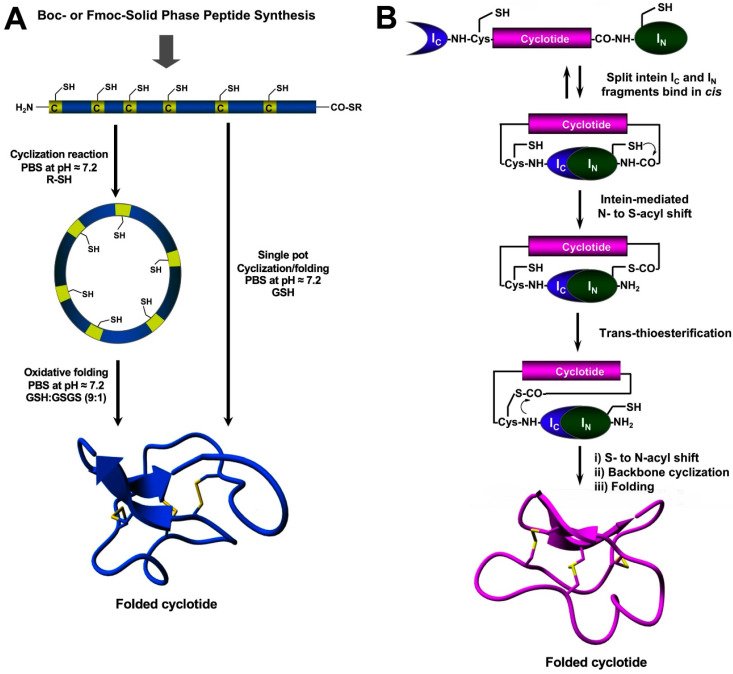
Different methods for the generation of native-folded cyclotides. (**A**) Chemical synthesis of cyclotides by employing an intramolecular version of native chemical ligation. This method requires the chemical production of a linear cyclotide precursor containing both an α-thioester moiety at the C-terminus and an N-terminal Cys residue. The linear precursor is then cyclized under reductive conditions and finally oxidatively folded using a proper redox buffer. The cyclization and oxidative folding reactions can be also efficiently performed in a ‘single pot’ reaction. This is accomplished by performing the cyclization in the presence of reduced GSH as the thiol cofactor. (**B**) Heterologous expression of cyclotides can be accomplished using protein trans-splicing (PTS) This approach has been employed for the generation of several MCoTI-cyclotides by using the native Cys residue located at the N-terminus of loop 6 to facilitate backbone cyclization. This method has been used to produce bioactive cyclotides using either eukaryotic or prokaryotic expression systems. Figure adapted from reference [23].

**Figure 5 molecules-27-06430-f005:**
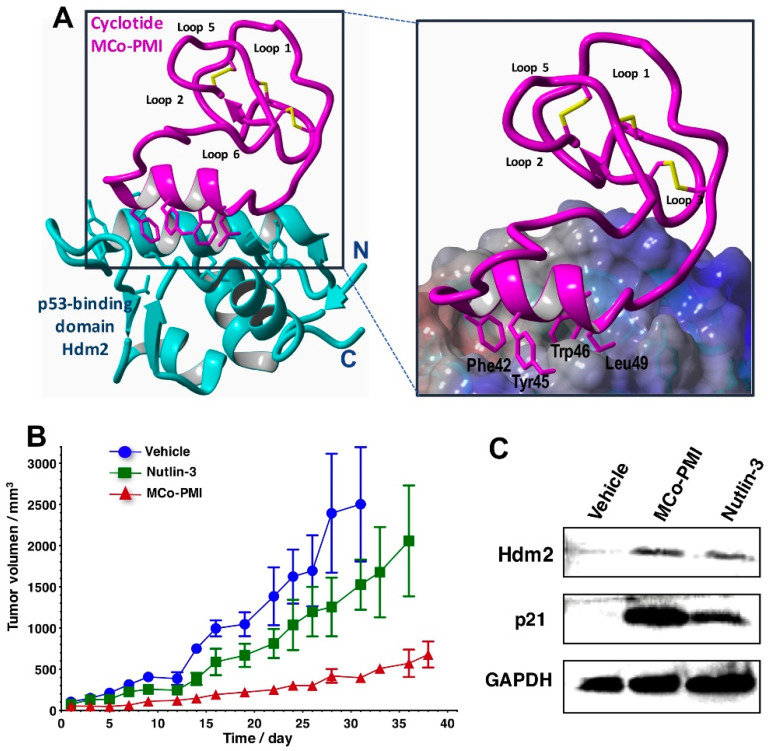
Structure and in vivo activity of an MCoTI-based cyclotide designed to antagonize an intracellular PPI [5]. (**A**) The structure of the bioactive cyclotide MCo-PMI (magenta) and its intracellular molecular target, the p53 binding domain of oncogene Hdm2 (blue), was determined in solution by NMR. Cyclotide MCo-PMI binds with low nM affinity to the p53-binding domains of Hdm2 and HdmX. (**B**) Cyclotide MCo-PMI activates the p53 tumor suppressor pathway and blocks tumor growth in a human colorectal carcinoma xenograft mouse model. HCT116 p53^+/+^ mouse xenograft models were treated with vehicle (5% dextrose in water), nutlin 3 (10 mg/kg) or cyclotide (40 mg/kg, 7.6 mmol/kg) by intravenous injection daily for up to 38 days. Tumor volume was monitored by caliper measurement. (**C**) Tumor samples were also subjected to SDS-PAGE and analyzed by western blotting for p53, Hdm2, and p21, indicating activation of p53 on tumor tissue in vivo. Figure adapted from reference [5].

**Table 1 molecules-27-06430-t001:** Engineered cyclotides from the trypsin inhibitor and Möbius subfamilies with novel biological activities. Table adapted and updated from references [11,23].

Cyclotide	Biological Activity	Loop	Application	Reference
**Modified**				
**Trypsin inhibitor subfamily**				
MCoTI-I	Antibacterial	6	Broad-spectrum antibacterial	[97]
MCoTI-I	MAS1 receptor	6	Lung cancer and myocardial	[35]
	agonist		infarction	
MCoTI-II	b-factor XIIa inhibitor	1 & 5	Antithrombotic	[98]
MCoTI-II	KLK4 inhibitor	1 & 8	Anti-cancer	[99]
MCoTI-I	CXCR4 antagonist	6	Anti-metastatic and anti-HIV PET-CT imaging	[33,34,66]
MCoTI-II	Antiangiogenic	5 & 6	Anti-cancer	[100]
MCoTI-II	SET antagonist	6	Potential anticancer	[101]
MCoTI-II	CTLA-4 antagonist	1, 3 & 6	Immunotherapy for cancer	[102]
MCoTI-II	BCR-Abl kinase	1 & 6	Chronic myeloid leukemia	[103]
	Inhibitor			
MCoTI-I	p53-Hdm2/HdmX	6	Anti-cancer	[5]
	Antagonist		pathway	
MCoTI-II	Tryptase inhibitor	1	Anti-cancer	[48]
MCoTI-II	Thrombospondin-1 (TSP-1)	6	Microvascular endothelial	[104]
	agonist		cell migration inhibition	
			anti-angiogenesis	
MCoTI-II	VEGF receptor agonist	6	Cardiovascular damage	[105]
			and wound healing	
MCoTI-II	β-Tryptase inhibitor	3, 5 & 6	Inflammation diseases	[106]
MCoTI-II	β-Tryptase inhibitor	1	Inflammation diseases	[107]
	Human elastase inhibitor			
MCoTI-II	FMDV 3C protease	1	Anti-viral for FMDV	[68]
	Inhibitor			
MCoTI-I	α-Synuclein-induced cytotoxicity inhibitor	6	Parkinson’s disease	[74]
**M** **öbius subfamily**				
Kalata B1	Immunomodulator	4	Protecting against multiple Sclerosis	[31]
Kalata B1	Immunomodulator	5 & 6	Protecting against multiple	[108]
			sclerosis	
Kalata B1	Neuropilin-1/2 antagonist	5 & 6	Inhibition of endothelial cell migration and angiogenesis	[109]
Kalata B1	Bradykinin B1 receptor antagonist	6	Chronic and inflammatory pain	[30]
Kalata B1	Melanocortin 4 receptor Agonist	6	Obesity	[110]
Kalata B1	Dengue NS2B-NS3 protease inhibitor	2 & 5	Anti-viral for Dengue virus infections	[111]
Kalata B1	VEGF-A antagonist	2, 3, 5 & 6	Anti-angiogenic, potential anti-cancer activity	[112]

## Data Availability

Not applicable.

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
