# Peer review of "Using the Cyclotide Scaffold for Targeting Biomolecular Interactions in Drug Development"

_molecules, 2022, doi:10.3390/molecules27196430_

Round 1

Reviewer 1 Report

The review article entitled “Using the cyclotide scaffold for targeting biomolecular interactions in drug development” by Jacob et al, provides an overview of the therapeutic applications of the cyclotide scaffolds in targeting biomolecular interactions. The authors describe the discovery and background of cyclotides and their applications in drug design in detail. Unfortunately, this review is overly broad in scope and I feel the current version of the manuscript falls short of insightful analysis. With minor revision, the article would be appealing to a wider range of audiences to this journal.

Below are some comments/suggestions on the flow of this article, references and some minor grammatical suggestions that should be addressed to improve the manuscript.

Major suggestions:

1.      The article is logically presented, however, I feel that the citations regarding cyclotides are not entirely up-to-date. For instance, there are some advancements in the synthesis, mutation and SAR studies of bracelet cyclotides being reported lately. I recommend the authors to update the references in the following section (Line 114-119, page 4) and briefly discuss how these recent advancements would impact the future application of bracelet cyclotides in drug development.

2.      Following the point above, the authors mentioned “Some bioactive cyclotides have also been shown to be orally active, although more detailed studies on the oral bioavailability of cyclotides may be required” (Line 504-505, page 14). Actually, the bioavailability of cyclotides was reported in 2019 (Poth et al., https://doi.org/10.1016/j.ijpharm.2019.05.001). Please cite the appropriate article and I recommend to include a brief discussion on the pharmacokinetics of cyclotides.

Minor technical corrections:

1.      Line 49: Ref 18 should also be cited for the statement “… therapeutic leads (see some recent reviews on the topic”.

2.      Figure 1, page 2: The beta-sheet display on the kalata B1 structure is not entirely correct.

3.      Line 205, page 7: They can be used “using”… Please delete “using” from this sentence.

4.      Line 403, page 12: This works highlights… Please change to “This work” highlights.

5.      Line 442, page 13: “…, the use of RNA-display technologies”. Please change it to “mRNA-display technologies”.

Author Response

Our detailed responses to the reviewer’s comments are the following:

Reviewer #1

1) The review article entitled “Using the cyclotide scaffold for targeting biomolecular interactions in drug development” by Jacob et al., provides an overview of the therapeutic applications of the cyclotide scaffolds in targeting biomolecular interactions. The authors describe the discovery and background of cyclotides and their applications in drug design in detail. Unfortunately, this review is overly broad in scope and I feel the current version of the manuscript falls short of insightful analysis. With minor revision, the article would be appealing to a wider range of audiences to this journal.

We thank the positive and encouraging comments from the reviewer, and we have modified it accordingly to make it more attractive to the reader (see below).

2) The article is logically presented; however, I feel that the citations regarding cyclotides are not entirely up-to-date. For instance, there are some advancements in the synthesis, mutation and SAR studies of bracelet cyclotides being reported lately. I recommend that the authors to update the references in the following section (Line 114-119, page 4) and briefly discuss how these recent advancements would impact the future application of bracelet cyclotides in drug development.

The paragraph has been updated as follows: “A recent report, however, has shown that introducing a single point mutation in loop 2, replacing Ile11 with either a Gly or Leu residue (Fig. 1), can substantially increase the folding yield in bracelet cyclotides [43]. This was demonstrated in several bracelet cyclotides containing an Ile residue in loop 2. This approach was successfully used to synthesize mirror image enantiomers and used quasi-racemic crystallography, allowing to elucidate of the first crystal structures of bracelet cyclotides [43]. This study should offer an alternative and efficient approach to obtaining bracelet cyclotides, facilitating easy access to their three-dimensional structures and providing a basis for further study of cyclotide structure and function and their future use as drug design scaffolds [43].”

And the following reference added accordingly (Ref.43): Huang, Y.H.; Du, Q.; Jiang, Z.; King, G.J.; Collins, B.M.; Wang, C.K.; Craik, D.J.Enabling efficient folding and high-resolution crystallographic analysis of bracelet cyclotides. Molecules 2021, 26(18): 5554.

2) Following the point above, the authors mentioned “Some bioactive cyclotides have also been shown to be orally active, although more detailed studies on the oral bioavailability of cyclotides may be required” (Line 504-505, page 14). Actually, the bioavailability of cyclotides was reported in 2019 (Poth et al., https://doi.org/10.1016/j.ijpharm.2019.05.001). Please cite the appropriate article, and I recommend including a brief discussion on the pharmacokinetics of cyclotides.

We have included the following pargraphs highlighting the report indicated by the reviewer as follows: “Oral activity has been described for cyclotide-containing traditional remedies and demonstrated in several kalata B1-grafted cylotides [29,37,38].  A recent study provided the first in vivo dose-exposure metrics for cyclotides using the prototypic cyclotide kalata B1 and two orally active kalata B1-grafted analogs, ckb-KAL and ckb-KIN [115]. This work determined the pharmacokinetic parameters in orally and intravenously dosed rats providing comparative pharmacokinetic parameters for natural and grafted cyclotides. The native and grafted cyclotides exhibited multiple compartment kinetics and measurable but limited oral bioavailability [115].

Preclinical pharmacokinetic evaluation is a crucial step in the progression of therapeutic candidates for use in a clinical setting, and more studies on bioactive cyclotides of other sub-families may be required. In addition, novel approaches to increase the oral bioavailability of cyclotides will have to be explored.

And the following reference added accordingly (Ref.115): Poth, A.G.; Huang, Y.H.; Le, T.T.; Kan, M.W.; Craik, D.J. Pharmacokinetic characterization of kalata b1 and related therapeutics built on the cyclotide scaffold. Int J Pharm 2019, 565, 437-446.

3) Line 49: Ref 18 should also be cited for the statement “… therapeutic leads (see some recent reviews on the topic”.

Done as requested.

4) Figure 1, page 2: The beta-sheet display on the kalata B1 structure is not entirely correct.

The beta-sheet on kalata b1 was defined automatically by the commercial molecular software package Yasara (http://www.yasara.org/) based on the pdb provided (pdb: 1NB1).

5) Line 205, page 7: They can be used “using”… Please delete “using” from this sentence.

Deleted as requested.

6) Line 403, page 12: This works highlights… Please change to “This work” highlights.

Corrected as requested.

7) Line 442, page 13: “…, the use of RNA-display technologies”. Please change it to “mRNA-display technologies”.

Changed as requested.

Reviewer 2 Report

The review of cyclotides for inhibition of protein-protein interactions is organized. Although the information are important for experts working on peptide-based derug development, there are many more comprehensive recent review articles that have been published and have not been cited here. 

Grover T, Mishra R, Bushra, Gulati P, Mohanty A. An insight into biological activities of native cyclotides for potential applications in agriculture and pharmaceutics. Peptides. 2021 Jan;135:170430. doi: 10.1016/j.peptides.2020.170430. Epub 2020 Oct 20. PMID: 33096195.

González-Castro R, Gómez-Lim MA, Plisson F. Cysteine-Rich Peptides: Hyperstable Scaffolds for Protein Engineering. Chembiochem. 2021 Mar 16;22(6):961-973. doi: 10.1002/cbic.202000634. Epub 2020 Nov 17. PMID: 33095969.

These reviews and others need to be cited and more imformation need to be provided what makes this review distinct from those that have been already published.

The cell-penetrating properties of cyclotides was briefly mentioned. Please provide more details and provide examples how they are uptaken by cells and whether they have been used in drug delivery.

Some information about the MIC values for some of these peptides could provide some informaiton about the potency.

Along the same line, some information about the toxicity of cylotides and range of concentration in normal cells will be useful when discussing the activity of compounds.

Table 1 can be more organized based on type of activity. All the peptides with similar activity should be next to each other.

Line 39 not clear: Change to " The use of highly constrainced cell-pereable polypeptides has recently...."

Author Response

Reviewer #2

1) The review of cyclotides for inhibition of protein-protein interactions is organized. Although the information are important for experts working on peptide-based drug development, there are many more comprehensive recent review articles that have been published and have not been cited here. 

The new references provided by the reviewer have been introduced in the introduction section (4th paragraph).

2) Some information about the MIC values for some of these peptides could provide some information about the potency.

The following information has been added: “A recent study reported the first design and synthesis of a novel MCoTI-based cyclotide with broad-spectrum antimicrobial activity in vitro against different ESKAPE pathogens (P. aeruginosa, S. aureus, K. pneumoniae, and E. coli), including 20 clinical isolates for the human pathogens P. aeruginosa and S. aureus. The median minimal inhibitory concentration (MIC ) 50% (MIC50) and MIC 90% (MIC90) values for several clinical strains of P. aeruginosa (n=20) were 1.5 μM and 3.1 μM, respectively; while for clinical isolates of S. aureus (n=20), the MIC50 and MIC90 were 6.25μM and 12.5μM, respectively [109]. The most active cyclotide also showed activity in vivo using a murine model of acute P. aeruginosa peritonitis [109].

2) Along the same line, some information about the toxicity of cylotides and range of concentration in normal cells will be useful when discussing the activity of compounds.

This is discussed in the review for different cases. In general, bracelet and Mobius cyclotides are quite toxic, while trypsin inhibitor cyclotides are practically non-toxic to mammalian cells. This is discussed in the manuscript, and the corresponding references are provided.

3) Table 1 can be more organized based on type of activity. All the peptides with similar activity should be next to each other.

Examples have now been grouped by activity; for example, all examples with anti-cancer activity have been grouped together for the different scaffolds.

Round 2

Reviewer 2 Report

The authors have addressed the comments.